# A Fiber Optic Ultrasonic Sensing System for High Temperature Monitoring Using Optically Generated Ultrasonic Waves

**DOI:** 10.3390/s19020404

**Published:** 2019-01-19

**Authors:** Jingcheng Zhou, Xu Guo, Cong Du, Chengyu Cao, Xingwei Wang

**Affiliations:** 1Department of Biomedical Engineering and Biotechnology, University of Massachusetts Lowell, 1 University Ave., Lowell, MA 01854, USA; Jingcheng_Zhou@student.uml.edu (J.Z.); Cong_Du@student.uml.edu (C.D.); 2Department of Electrical and Computer Engineering, University of Massachusetts Lowell, 1 University Ave, Lowell, MA 01854, USA; Xu_Guo@student.uml.edu; 3Department of Mechanical Engineering, University of Connecticut, Storrs, CT 06269, USA; chengyu.cao@uconn.edu

**Keywords:** fiber optic sensor, high temperature monitoring, ultrasonic, photoacoustic, Fabry-Perot

## Abstract

This paper presents the design, fabrication, and characterization of a novel fiber optic ultrasonic sensing system based on the photoacoustic (PA) ultrasound generation principle and Fabry-Perot interferometer principle for high temperature monitoring applications. The velocity of a sound wave traveling in a medium is proportional to the medium’s temperature. The fiber optic ultrasonic sensing system was applied to measure the change of the velocity of sound. A fiber optic ultrasonic generator and a Fabry-Perot fiber sensor were used as the signal generator and receiver, respectively. A carbon black-polydimethylsiloxane (PDMS) material was utilized as the photoacoustic material for the fiber optic ultrasonic generator. Two tests were performed. The system verification test proves the ultrasound sensing capability. The high temperature test validates the high temperature measurement capability. The sensing system survived 700 °C. It successfully detects the ultrasonic signal and got the temperature measurements. The test results agreed with the reference sensor data. Two potential industry applications of fiber optic ultrasonic sensing system are, it could serve as an acoustic pyrometer for temperature field monitoring in an industrial combustion facility, and it could be used for exhaust gas temperature monitoring for a turbine engine.

## 1. Introduction

Temperature is one of the most critical parameters in industry and science. In some applications, temperature sensors which are immune to electromagnetic interference and display durability to harsh environments, remote sensing capability, multiplexing capability, wide operating range, and allow long-distance interrogation without an electrical interface are required. Fiber optic sensors provide a good solution for many of these challenges. The concept of using fiber optic techniques for temperature sensing purposes was first discussed fifty years ago, and what would now be recognized as fiber optic sensors were introduced into the market.

Many fiber optic temperature sensors have been designed and built in the past decades, due to their advantages of miniature design, electromagnetic immunity, excellent stability, and enable operation in hazardous environments, and so on. Fiber optic temperature sensors with Fiber Bragg Grating (FBG) and Long Period Fiber Grating (LPFG) inscribed in optical fibers are proposed [1,2,3,4], fiber optic temperature sensing schemes based on the use of fluorescence lifetime decay detection are also demonstrated [5,6]. Some other types of fiber optic temperature sensors are also proposed, which mainly based on Mach-Zehnder interferometer [7], and Michelson interferometer [8]. Besides the regular optical fiber, special optical fibers, such as hollow-core fiber [9], photonic crystal fiber [10], no-core fiber [11] and sapphire fiber [12], are used to enhance the temperature measurement sensitivity or measurement range. Brillouin scattering and low-coherence interferometry are used to for the distributed fiber-optic temperature systems [13,14].

In some applications, such as temperature field monitoring in coal-fired boilers or exhaust gas temperature monitoring for turbine engines, industry demands a 2D or 3D temperature distribution profiler. Traditionally, the industry is using electronic transducers as the acoustic pyrometers for the temperature field monitoring. So that they can get the temperature information of a line. The 2D or 3D temperature distribution profiler can be reconstructed by using multiple temperature information lines. However, these electronic transducers have some drawbacks. They cannot survive in the boiler high temperature environment, and also have electromagnetic interference. In this paper, the first active non-contact all-optical fiber sensing system is presented. Since this system is fabricated using optical fibers, it can survive in a higher temperature than the traditional electronic transducers. Also, the fiber optic ultrasonic sensing system features immunity to electromagnetic interference, high sensitivity, and small size. They are especially suitable for applications in harsh environments.

A fiber optic ultrasonic generator was used as a signal generator in this system. This ultrasonic generator is based on the photoacoustic (PA) principle which converts the light energy to an acoustic signal. Researchers have developed a variety of materials as the photoacoustic material in the past ten years. An ideal PA material should feature a high optical energy absorption capability and a high coefficient of thermal expansion (CTE). Metal films are first used as photoacoustic materials due to easy fabrication and high light absorption [15]. However, the photoacoustic conversion efficiency of metals is low since their low thermal expansion. Researchers have developed composite materials with both high thermal expansion and light absorption, such as gold-nanocomposites [16,17,18], carbon black combined with PDMS (black PDMS) [19,20], CNTs [21], candle soot nanoparticles [22], and polymer-thin metal-polymer [23]. In this paper, black PDMS material was used as a photoacoustic material. 

A Fabry-Perot (FP) fiber sensor was used as a signal receiver [24,25]. This FP fiber sensor receiver is based on FP interferometer principle. Fiber based Fabry-Perot sensors are known for decades, many of them are high temperature stable and can be used for acoustic detection [26,27,28,29]. In this paper, a new FP fiber sensor was designed and built for the system. 

This work has great significance because the fiber optic sensor system can survive high temperatures (up to 700 °C) and the optically generated acoustic signals can measure even higher temperature where the fibers do not reach (e.g., 1500 °C). In the future, the 2D and 3D temperature distribution profile will be reconstructed by using a recursive algorithm based on Gaussian radial basis functions (GRBF) parameterization.

This paper is organized as follows: Section 2 presents the methodology. This section includes the design and development of the fiber optic ultrasonic generator and FP fiber sensor, and the principle of time-of-flight (TOF) temperature measurement method. Section 3 describes the verification for the proposed sensing system. Section 4 describes the high temperature measurement test. Section 5 concludes the paper.

## 2. Methodology

### 2.1. Fiber Optic Ultrasonic Generator

The fiber optic ultrasonic generator is based on the photoacoustic (PA) principle, which uses optical signals to generate ultrasonic waves (Ultrasonic waves are acoustic waves that the frequency greater than 20 kHz). Applying photoacoustic involves two major steps: (1) Conversion of optical energy to thermal energy and (2) Generation of ultrasonic signal due to thermal expansion effect [30,31,32,33,34,35].

The fiber optic ultrasonic generator converts laser pulse energy, incident on a photoacoustic thin film into ultrasonic waves. The fiber optic ultrasonic generator is easy to fabricate. Black PDMS (20% carbon black + 80% PDMS) was used as the photoacoustic material. For black PDMS fabrication, a PDMS silicone elastomer kit (Sylgard 184, Dow Corning, Midland, MI, USA) and a carbon black (Conductex Sc Vara, Birla Carbon, Marietta, GA, USA) are used for the fabrication. The carbon black and the PDMS matrix were mixed by a speed mixer (SpeedMixer™ DAC 150 FVZ, FlackTek Inc., Landrum, SC, USA) at 2000 rpm of 1 min for 10 times. The upper layer suspension of the mixer were coated on glass slides and rested overnight.

In this paper, a glass slide was coated with black PDMS. Light launched from a Surelite I-10 532 nm Nd:YAG nanosecond laser (Continuum, San Jose, CA, USA) traveling through a 1000/1035 µm optical fiber was shone onto the black PDMS and the ultrasonic signal was thus generated. Figure 1 shows the structure of the fiber optic ultrasonic generator.

Figure 2 shows an ultrasonic signal that was generated by the fiber optic ultrasonic generator. The measurement was performed in water. A fiber optic ultrasonic generator was used as the signal generator. A hydrophone (HGL-0200, Onda, Sunnyvale, CA, USA) was used as the signal receiver. The distance between the generator and the receiver was 1 mm. The peak to peak amplitude of the ultrasonic pressure was measured to be 0.11 V. Since the hydrophone used in this experimental was 50.00 nV/Pa. The pressure can be converted to 2.20 MPa. The pulse width was measured as 160 ns. After performing the Fourier transform, the bandwidth of the frequency range was at least 20 MHz as shown in Figure 2b [17].

### 2.2. Fabry–Pérot (FP) Fiber Sensor

The structure of the Fabry-Perot (FP) fiber sensor is shown in Figure 3a. A quartz coverslip, an aluminum plate, a ferrule and a single mode fiber (SMF) were used to build this Fabry-Perot sensor. A beam of laser light was launched into the SMF, partially reflected from the tip of the SMF. The transmitted light was reflected from coverslip. These two beams of reflected light form the FP interference. The ultrasonic signal impinging on the coverslip changed the distance between the coverslip and the SMF, and then changed the interference signal.

The sensitivity of the sensor defines that how much the center of the diaphragm will be deformed when a certain acoustic pressure applied on it. The following equation defines it [36]:
(1)$$Y_{c}=\frac{3\left(1-{µ}^{2}\right){\left(d/2\right)}^{4}}{16{\mathrm{Eh}}^{3}}\cdot 10^{9}$$

E is the quartz’s Young’s modulus, E = 7.2 × 10^10^ Pa; µ is the quartz Poisson ratio, µ = 0.17; h is the thickness of the quartz coverslip, h = 0.10 mm; d is the diameter of the aluminum hole, d = 2.54 mm. Y_c_ = 0.0032 nm/Pa.

The sensitivity of the FP fiber sensor on the optical sensing analyzer:
(2)$$S_{\mathrm{CTS}}=1.55\times \frac{Y_{c}}{I}$$
where I is the FP cavity length (I = 3 μm), $S_{\mathrm{CTS}}$ was calculated as 1.6 × 10^−3^ nm/Pa, which means that, with 1 Pa pressure, the spectrum would shift 1.6 × 10^−3^ nm.

In sound applications, a resonant frequency is a natural frequency of vibration determined by the physical parameters of the vibrating object. The resonant frequency of the FP fiber sensor was determined by [37]:
(3)$$f_{00}=\frac{\alpha_{00}}{4\pi }{\left[\frac{E}{3w\left(1-{µ}^{2}\right)}\right]}^{1/2}\left[\frac{h}{\;{\left(d/2\right)}^{2}}\right]$$
where f_00_ is the lowest resonant frequency; α_00_ is a constant related to the vibrating modes, which is 10.21 for the lowest natural frequency; w is the mass density of the quartz w = 2.50g/cm^3^. For our experiment f_00_ was calculated as 0.19 MHz.

Since this FP fiber sensor will be used in very high temperature environment. The packaging of this FP fiber sensor is important. The packaging of this FP fiber sensor is shown in Figure 3b. A 4.76 mm outside diameter copper tube was used in this packaging. The diameter of the aluminum plate Part A hole was 2.54 mm. The diameter of the aluminum plate Part B hole was 5.08 mm. The 4.76 mm copper tube could be inserted into the aluminum plate Part B. The aluminum adhesive was used to seal the ferrule with the single mode fiber, the ferrule with the copper tube, as well as the copper tube with the aluminum plate. 

### 2.3. Time-of-Flight Temperature Measurement Method

The principle of using time-of-flight (TOF) method to measure the temperature of a medium is straightforward. The speed of sound can be correlated with the temperature of a medium in which the ultrasonic wave travels. Therefore, if the time-of-flight (TOF) of an ultrasonic pulse between two fixed points within the medium can be known, one can calculate the speed of sound within the medium, which will lead to the temperature along that path within the medium. The temperature T is governed by:
(4)$$T={\left(\frac{c}{B}\right)}^{2}$$
where c is the sound velocity and B is the acoustic constant of the air. Sound velocity will be calculated by the flight time of an acoustic wave divided by the flight distance [24,25].

## 3. Fiber Optic Ultrasonic Sensing System Verification

### 3.1. Experimental Setup

A fiber optic ultrasonic sensing system verification experiment was performed to evaluate the performance of the system. The experiment was held at 20 °C. A fiber optic (ultrasonic) generator was used as the signal generator. Black PDMS material was coated on a glass slide. Light launched from the 532 nm Nd:YAG nanosecond laser traveled through a 1000/1035 µm optical fiber and was shone onto the black PDMS and this generated the ultrasonic signal. A microphone (TMS 130C21, PCB, Depew, NY, USA) and an FP fiber sensor (V20161202TEST2) were used as the signal receivers, as shown in Figure 4a,b, respectively. The distances between the generator and receiver were set as 10, 20 and 30 mm, respectively.

For the microphone receiver, a power supply (482A06, PCB, Depew, NY, USA) was used for the microphone. The output electrical signal from the microphone was recorded by a data acquisition card (DAQ) (M2i.4032, Spectrum, Hackensack, NJ, USA) at a sampling rate of 50 MHz. For the FP sensor receiver, a tunable laser (TLB-6600, Venturi^TM^ Tunable Laser, Santa Clara, CA, USA) was used as a light source to excite the FP fiber sensor through a circulator. The output power for the tunable laser was set as 7.60 mW. The reflected light through the circulator was detected by a photodetector (PDA10CS, Thorlabs, Newton, NJ, USA), which converted the optical signal into electrical signal. The photodetector was set at 40 dB. The spectrum of the FP fiber sensor is shown in Figure 5. The wavelength of the FP fiber sensor was set between 1557 nm to 1557.5 nm (between these wavelengths, the slope of the waveform was bigger which means a higher sensitivity). The DAQ recorded the output electrical signal from photodetector at a sampling rate of 50 MHz.

### 3.2. Results and Discussion

The ultrasonic signal results are shown in Figure 6. Figure 6a is the ultrasonic signal detected by the microphone. Figure 6b is the ultrasonic signal detected by the FP fiber sensor. We had done some background noise reduction on the original signal. We first found the baseline of the original signal, then we used the original signal to subtract the baseline, then we smoothed the signal. The travel time we got was the time difference between the signals sent from the generator to the signal detected by the receiver. The time 0 was defined as the signal transmitted from the generator. The time-of-arrival of the signal was the time at the start point of the ultrasonic signals. It’s near the first peak of the ultrasonic signal. It was the starting point for a big variation in the signal. The travel time was 30.12 µs, and 30.20 µs detected by the microphone and the FP fiber sensor as shown in Figure 6. 

Since the distances between generator and receiver was 10 mm. The speed of sound v was calculated as:
(5)$$v=\frac{s}{t}$$

The speed of sound v calculation result was 331 m/s and 332 m/s, respectively. The speed of sound in the air was 343 m/s at 20 °C [38]. This agreed with the experiment results.

At the distance of 10 mm, the peak to peak voltage value (Vpp) from the microphone and the FP sensor were 5 mV and 1.20 mV, respectively. The microphone had a 4.2 times of Vpp than the FP fiber sensor. We assumed the microphone also had a 4.2 times of sensitivity than the FP fiber sensor. The sensitivity of the microphone was 22.51 mV/Pa. Therefore, the sensitivity of the FP fiber sensor was calculated as 5.3 mV/Pa.

The gain (at 40 dB) for the photodetector was 0.75 × 10^5^ V/A, the responsivity for the photodetector was 1 A/W between 1557 nm to 1557.5 nm wavelength. The sensitivity of the FP fiber sensor was converted as 7 × 10^−8^ W/Pa. 

The output power for the tunable laser was 7.60 mW, the power at P_1_ (−19 dB) and P_2_ (−20 dB) were 0.096 mW and 0.076 mW. 

The slope a of the FP fiber sensor between 1557 nm to 1557.5 nm wavelength was calculated as:
(6)$$a=\left|\frac{p_{1}-p_{2}}{\lambda_{1}-\lambda_{2}}\right|$$

From Figure 5, $\lambda_{1}$, $\lambda_{2}$ were 1557 nm and 1557.5 nm. The slope calculation result was 0.04 mW/nm. The sensitivity of the FP fiber sensor on the optical sensing analyzer was converted to 1.7 × 10^−3^ nm/Pa. Therefore it matched the calculation results in Section 2.2, which $S_{\mathrm{CTS}}$ was 1.6 × 10^−3^ nm/Pa.

After performing the Fourier transform, Figure 7 shows the 10 mm distance test ultrasonic signal in the frequency domain. There was a peak at 0.19 MHz. Therefore it matched the resonant frequency calculation results in Section 2.2, which the resonant frequency was 0.19 MHz.

## 4. Fiber Optic Ultrasonic Sensing System Verification

### 4.1. Experimental Setup

Four fiber optic ultrasonic sensing system high temperature tests with different temperature (100, 300, 500 and 700 °C) were performed to evaluate the high temperature measurement capability of the system. In this paper, 700 °C high temperature test setup is shown in Figure 8. A fiber optic (ultrasonic) generator was used as the signal generator. A black PDMS glass coverslip was attached to the water block of the water cooling system. The water cooling system was used in this test for cooling the black PDMS. Light launched from the Surelite I-10a 532 nm Nd:YAG nanosecond laser traveled through a 1000/1035 µm optical fiber and was shone onto the black PDMS and generated the ultrasonic signal. An FP fiber sensor (V20170321TEST1) was used as the signal receiver. A TLB-6600 tunable laser (TLB-6600, Venturi^TM^ Tunable Laser, Santa Clara, CA, USA) was used as a light source to excite the FP fiber sensor through a circulator. The output power of the tunable laser was set as 7.60 mW. The reflected light through the circulator was detected by a Thorlabs PDA10CS photodetector (PDA10CS, Thorlabs, Newton, NJ, USA), which converted the optical signal into electrical signal. The photodetector was set as 30 dB. The spectrum of the FP fiber sensor is shown in Figure 9. The wavelength of tunable laser was set at 1565.7 nm. This FP fiber sensor (V20170321TEST1) has the same structure as the FP fiber sensor (V20161202TEST2), but the spectrum is slightly different. The FP cavity lengths and the thicknesses of the wafers were slightly different which caused the spectrum difference. The distance between the generator and the receiver was fixed at 10 mm. When the laser source released a pulsed signal, a trigger signal was sent from the laser system to trigger an M2i.4032 data acquisition card (M2i.4032, Spectrum, Hackensack, NJ, USA) at a sampling rate of 50 MHz. The distance between the generator and the receiver was fixed at 10 mm. Temperature between the generator and the receiver was recorded by a thermocouple (KHXL-116G-RSC-24, OMEGA, Norwalk, CT, USA). The door of the furnace was covered with aluminum foils for keeping the high temperature inside the furnace. The furnace temperature was set from room temperature (24 °C) to high temperature (700 °C).

### 4.2. Results and Discussions

From Figure 10, it can be inferred that there were clear ultrasonic signals when the furnace was set to temperatures of 24 °C (room temperature) and 700 °C (high temperature), respectively. The FP fiber sensor spectrum became stable at 700 °C after 20 min when the furnace reached its setting temperature. At 700 °C, the spectrum of the FP fiber sensor had a difference compared with that under room temperature as shown in Figure 9, that’s caused by the bonding glue. But, the spectrum became stable after 20 min at 700 °C, so that we can use that spectrum for the ultrasonic signal detection.

The temperature based on the optical system was calculated and one of the results is shown in Figure 10. Since the distance between generator and receiver was not exactly 10 mm. The real distance s_1_ has been calculated by multiplying the speed of sound v_1_ with the travel time t_1_ at 24 °C room temperature:
(7)$$s_{1}=v_{1}t_{1}$$

The speed of sound is 345.549 m/s at 24 °C [38]. The real distance s can be calculated as 9.413 mm. Then we use the real distance s_1_ to divide the travel time t_2_ at setting 700 °C to calculate the speed of sound at setting 700 °C:
(8)$$v_{2}=\frac{s_{1}}{t_{2}}$$

The speed of sound at setting 700 °C can be calculated as 559.631 m/s. It represented 506.25 °C according to the temperature and speed equation calculator [38]. The travel time results and the furnace temperature setting, the thermocouple reference temperature are listed in Table 1. Data was recorded three times at 700 °C test.

Other three fiber optic ultrasonic sensing system high temperature tests (100, 300, 500 °C temperature test) data and this 700 °C test data are plotted in Figure 11. The biggest temperature variation over the measurement is 2.64%. The furnace setting temperature, reference thermocouple temperature and the temperature calculated by travel time are different, because: (1) the temperature distribution was not consistent between the generator and the receiver; (2) as the aluminum foil covered the furnace door, the reference temperature measured by the thermocouple in the middle between the generator and the receiver cannot represent the real temperature; (3) The furnace temperature setting sensor was fixed on the furnace wall and thus cannot represent the testing area.

The relationship between travel time and thermocouple reference temperature is shown in Figure 12. A linear fitting line is plotted in the figure.

## 5. Conclusions

In this paper, we have designed, fabricated, and characterized the fiber optic ultrasonic sensing system to measure high temperature in air condition. This system is the first active non-contact all optical fiber sensing system using optically generated acoustic signals to operate in the high temperature harsh environment. It based on PA generation technique by using black PDMS as the ultrasound generation material. It also based on Fabry-Perot principle by using FP cavity as the signal receiver part. The verification experiment was performed to validate the sensing capability. The experimental results showed that ultrasonic signals can be detected by the system. 

The fiber optic ultrasonic sensing system high temperature tests were performed to validate the high temperature measurement capability. The results showed that the novel fiber optic ultrasonic sensing system could work at 700 °C. It has the potential to be used in high temperature environments. The system survived in high temperature environment (700 °C) for at least 3 hours, and it’s still workable. The maximal and minimal distance between the generator and reviver is 1 mm to 50 mm. If we replaced the FP fiber sensor to a microphone, the maximal measurement distance could be increased to 1000 mm. In summary, the fiber optic ultrasonic sensing system could lead to the development of a new generation temperature sensor for temperature field monitoring in coal-fired boilers or exhaust gas temperature monitoring for turbine engines.

## Figures and Tables

**Figure 1 sensors-19-00404-f001:**
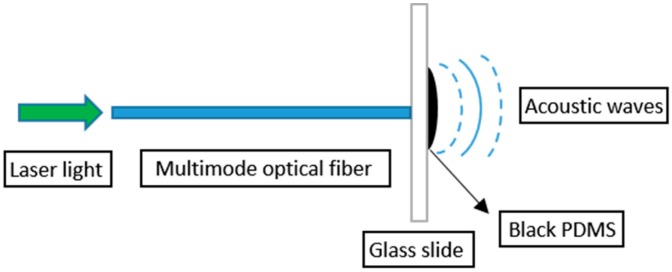
The structure of fiber optic ultrasonic generator.

**Figure 2 sensors-19-00404-f002:**
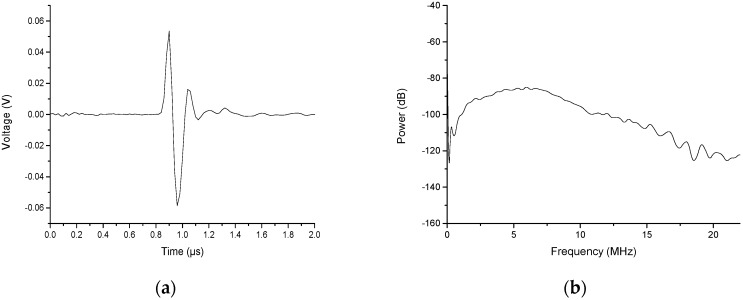
The ultrasonic signal generated by the fiber optic ultrasonic generator. (**a**) The profile of a generated ultrasonic signal. (**b**) The frequency domain of the generated ultrasonic signal.

**Figure 3 sensors-19-00404-f003:**
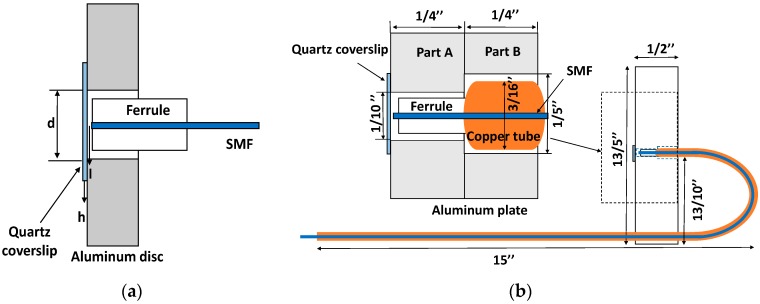
The FP fiber sensor. (**a**) The structure of an FP fiber sensor. (**b**) The packing of an FP fiber sensor.

**Figure 4 sensors-19-00404-f004:**
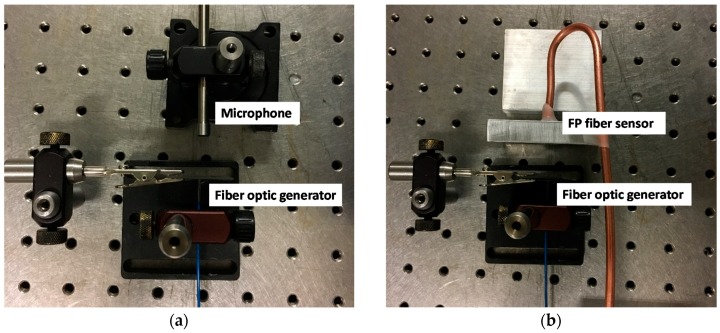
Experimental setup for the fiber optic ultrasonic sensing system verification. (**a**) The fiber optic generator and microphone system. (**b**) The fiber optic generator and FP sensor system.

**Figure 5 sensors-19-00404-f005:**
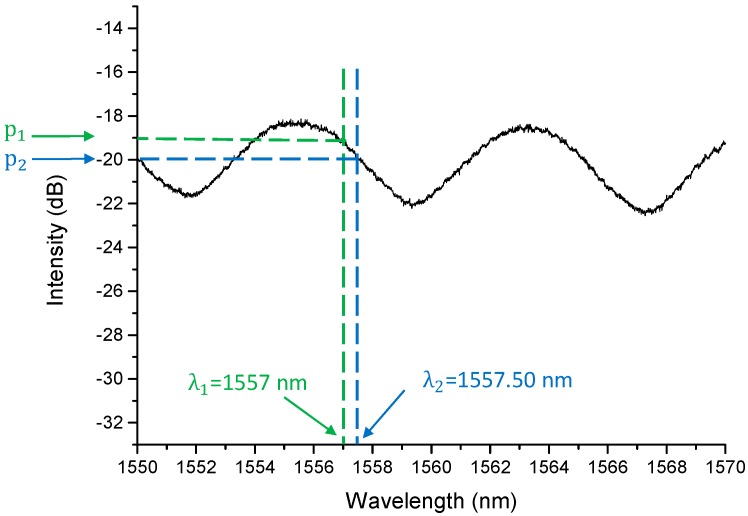
The spectrum of the V20161202TEST2FP fiber sensor.

**Figure 6 sensors-19-00404-f006:**
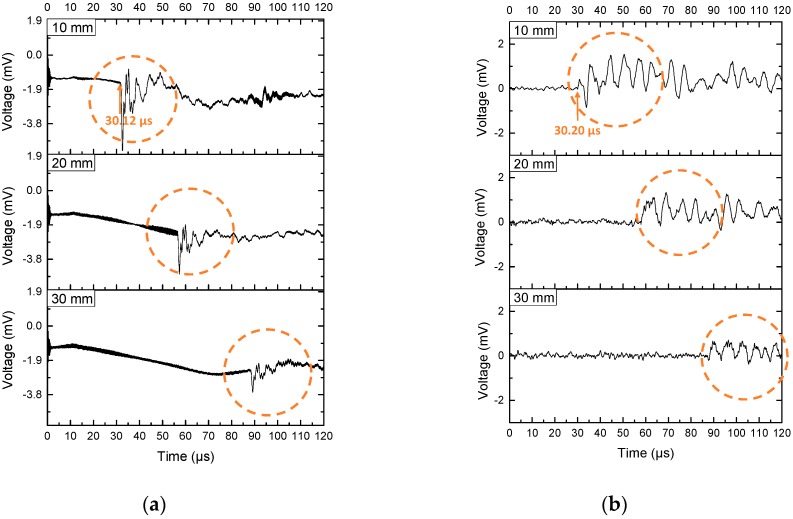
The ultrasonic signal detected by the (**a**) microphone and (**b**) FP fiber sensor.

**Figure 7 sensors-19-00404-f007:**
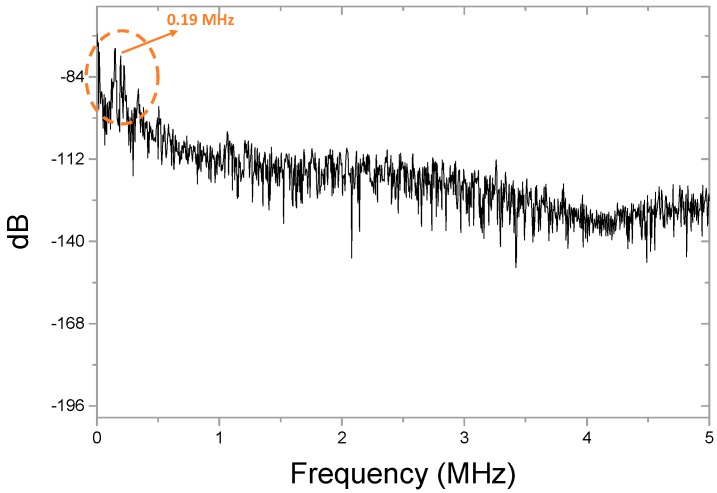
The frequency domain of the signal from the FP fiber sensor (10 mm distance test).

**Figure 8 sensors-19-00404-f008:**
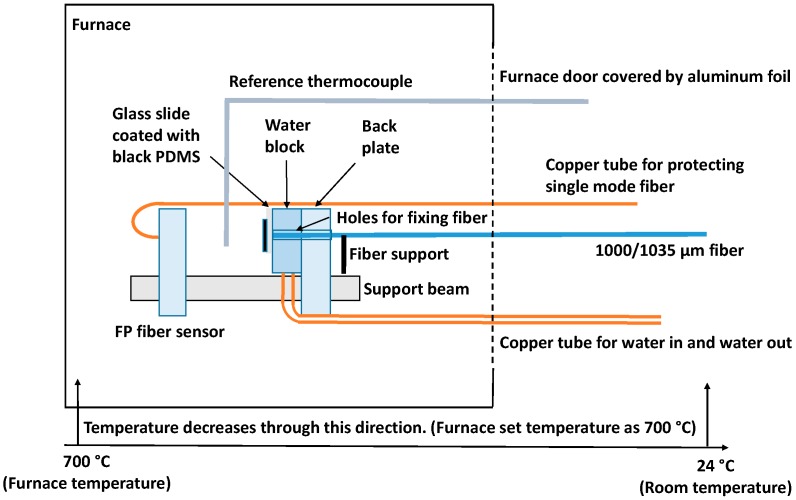
Experimental setup for the fiber optic ultrasonic sensing system high temperature test.

**Figure 9 sensors-19-00404-f009:**
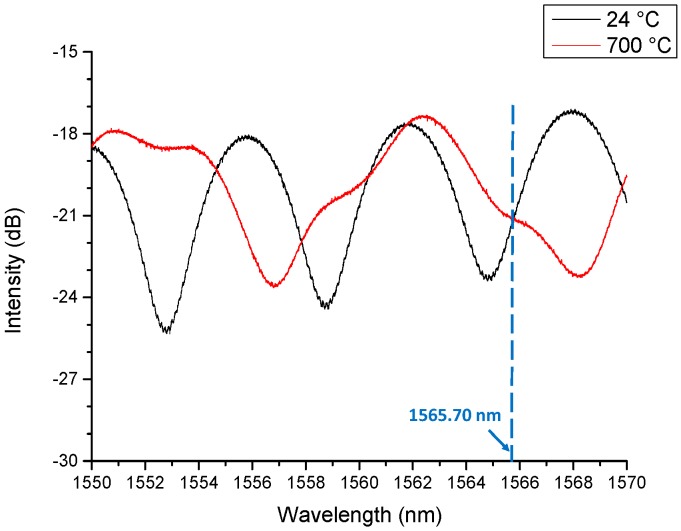
The spectra of the V20170321TEST1FP fiber sensor at different temperatures.

**Figure 10 sensors-19-00404-f010:**
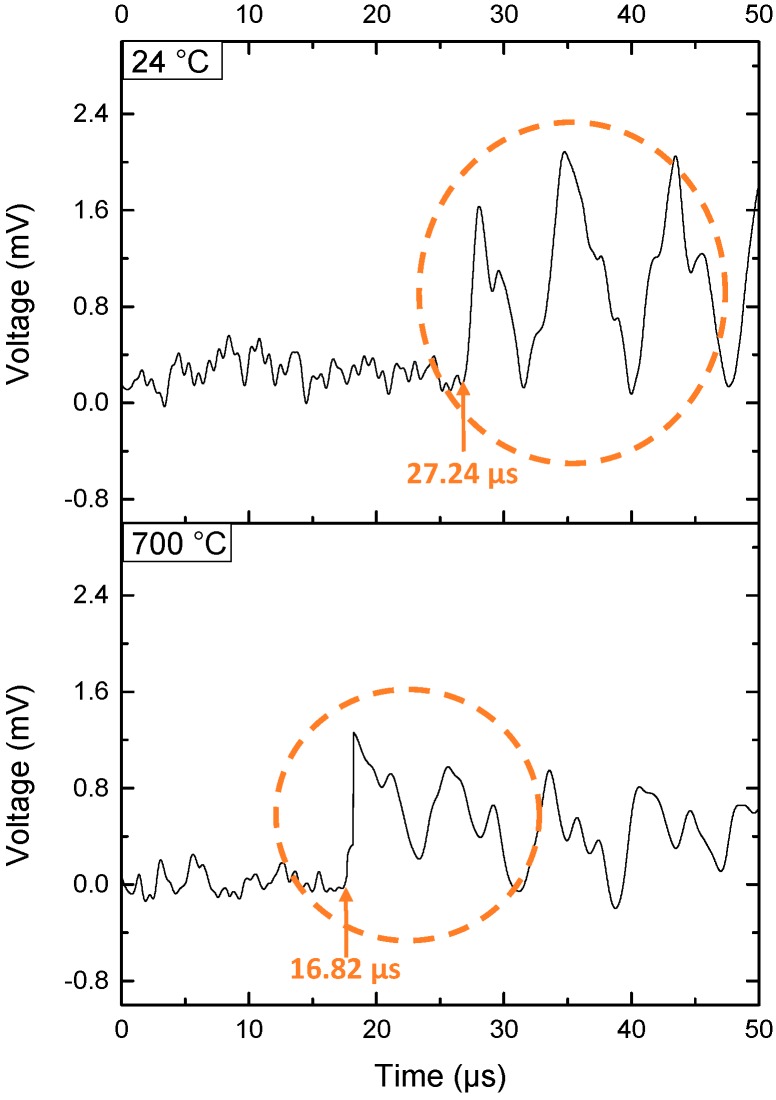
Ultrasonic signals for 700 °C high temperature test.

**Figure 11 sensors-19-00404-f011:**
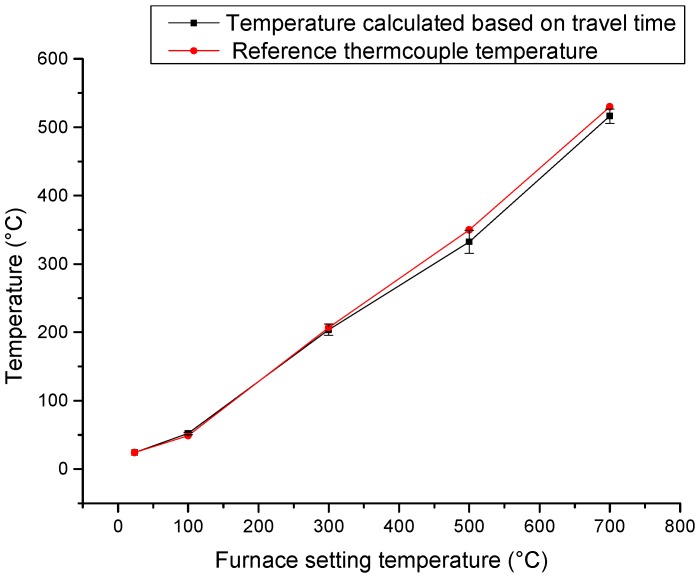
Thermocouple reference temperature compared with temperature calculated based on travel time at the same furnace temperature setting.

**Figure 12 sensors-19-00404-f012:**
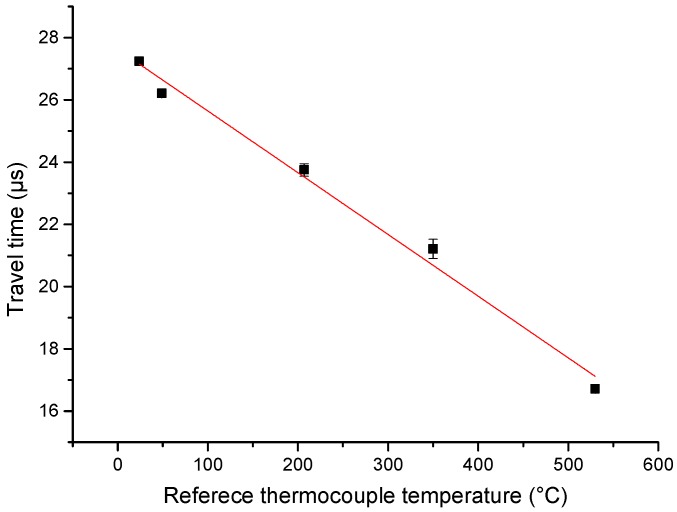
The relationship between travel time and thermocouple reference temperature.

**Table 1 sensors-19-00404-t001:** The relationship between different temperature results.

Furnace Setting Temperature (°C)	Temperature Reading between the Generator and the Receiver from a Thermocouple (°C)	Travel Time from Our Optical System (µs)	Temperature Calculated Based on the Travel Time (°C)
24	24	27.24	24
700	530	16.82	506.25
700	530	16.72	515.60
700	530	16.60	527.05

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
