# Peer review of "A Fiber Optic Ultrasonic Sensing System for High Temperature Monitoring Using Optically Generated Ultrasonic Waves"

_sensors, 2019, doi:10.3390/s19020404_

Round 1

Reviewer 1 Report

The authors present the design and evaluation of high-temperature capable temperature sensor based on photoacoustic generation and a Fabry-Perot based microphone. The authors describe the fabrication and verification of the sensor response. Generally, the sensor would be interesting to the measurement community, however significant revisions of the manuscript are required to better explain the measurement device and to evaluate its performance. Comments are provided below.

1.       The review has a disagreement with one of the arguments of the introduction. The authors state that often distributed (2D or 3D) temperature information is needed for certain applications. This is certainly true. The authors then state that sensors such as Bragg gratings cannot provide this distributed information and that the current sensing system can. The reviewer disagrees with this on several points: first, there are many examples of distributed fiber-optic temperature systems, for example using Brillouin scattering or low-coherence interferometry. These are not mentioned in the introduction. Second, while Bragg gratings do measure point values, they can be closely spaced along a fiber and wavelength multiplexed, providing near-distributed sensing. The current sensor proposed by the authors, on the other hand, cannot be closely spaced along an optical fiber for high density multiplexing. Therefore it does not provide a solution for distributed sensing.

2.       The data plotted in figure 1 is not clear. What measurement was performed? How was this signal measured? What was the input signal to the photoacoustic generator? How close do they match? No description of test setup is provided. I am asking because it is important to know other conversion factors for ultrasonic wave into voltage and how close the ultrasonic wave is to the desired input.

3.       How do the authors mathematically define the time-of-arrival of the signal? This would have an effect on the accuracy of the temperature measurement. As many of the signals become distorted, a robust definition is needed.

4.       It is not clear to the reviewer what the different tests in figure 9 represent. Were these measurements at different locations in the furnace? Or where they different trials at same locations? The paragraph above the figures says there were "differences in measured data”? Does this mean between tests or between the reference and test?

5.       The test data in figure 9 does not really present a satisfactory evaluation of the performance of the sensor. Why was the maximum temperature of the furnace not varied and the measurements performed at a many temperatures? Then the calibration curve for the sensor could be derived and also compared to the theoretically predicted curve. Finally, estimates of the measurement accuracy and precision could be made. The reviewer thinks that these data are critical to the presentation of the scientific quality of the manuscript and are therefore necessary to be included.

Reviewer 2 Report

In the manuscript the authors presented fiber tip based Fabry-Perot sensor in combination with ultrasonic wave generator for temperature sensing. The idea is to measure speed of ultrasonic wave propagation from fiber ultrasonic source to Fabry-Perot sensor and calculate temperature from this information. Design, fabrication and testing of the sensor was presented. 

Although the work includes some novelty and such sensors can be of interest for the community, the manuscript lacks scientifically sound presentation and keeps many questions open. Hence major revisions might be needed to bring the manuscript to the shape suitable for publication. Below some major and minor comments from my side. 

- Introduction section much deal more about fiber based Fabry-Perot sensors, rather than present general overview of different temperature sensors. Fiber based Fabry-Perot sensors are known for decades, many of them are high temperature stable and can be used for acoustic detection as well. There are commercialized solutions as well. So authors should better position their work in this large domain with hundreds of published works on fiber Fabry-Perot sensors, rather than presenting general fiber sensing approaches.

- In the introduction authors talks (also in the abstract) about distributed sensing, 2D and 3D sensing, while it is not clear how their sensor can address such problems. Needs to be explained and demonstrated well or they should skip mentioning such possibilities. 

- Some derivations in Section 2.2 seems to be not connected with the story of the experiments. Comparison of theoretical and experimental sensitivities is not presented as well. 

- Section 2.3 needs more substantiation. What is time and distance measurement error and its influence on the temperature measurement?

- Last general comment is about the conclusions of the general performance and prospects of the sensor. Apart of the experimental results no deeper analysis of the sensor specifications and performance is presented such as: What is the limit of detection? How accurate is the sensor? How repeatable is the sensor? What is maximal and minimal distance between the acoustic generator and the sensor head?

Some other minor comments:

- Section 2.2 - references are needed for the formulas. 

- Figure 1 - explain how did you measure this?

- Use either millimeters or inches (better the first one).

Author Response

Please see the attached report Notes.

Reviewer 3 Report

Authors present the design of a system for temperature sensing. The system is composed by 2 stages, where the first one is an optical fiber ultrasonic generator and the second stage is a fiber the Fabry-Perot (FP) fiber sensor which monitors the ultrasonic waves. According to the authors the sensing system is able to detect the temperature of the air between the ultrasonic generator and the FPI sensor. In general the sensing arrangement is interesting however there are some aspects that require further discussion and descriptions, particularly regarding on the ultrasonic generator stage. Hence I suggest to consider it for publication after major revisions are carried out by authors. Some specific aspects that I consider should be addressed by authors are the following.

1.       In section 2.1 regarding on the fiber optic ultrasonic generator, it is necessary that authors include a figure of the implemented setup (probably as Fig. 1a). Moreover, it is necessary to provide further information about the fabrication process describing with detail how the glass slide was coated with PDMS.

2.       About the ultrasonic generation it is necessary to clarify the principle of operation, in particular authors mentioned that the laser that was used as a light source provided nanosecond pulses, however it is not clear, for my point of view the characteristics of the generated ultrasonic wave. Here it is necessary to clearly describe the relationship between the frequency and power of the laser pulses and the frequency and amplitude of the generated ultrasonic waves.

3.       Regarding to the FPI sensor, it is necessary to define if there a minimum “intensity” of the ultrasonic wave in order to be detected or it only dependent on the frequency of the ultrasonic wave.

4.       Equation (1) defines how much the center of the diaphragm will be deformed as a function of the acoustic pressure applied on it (nm/Pa), that in this case is Yc = 0.0032 nm/Pa. Afterwards equation (2) helps to calculate the lowest resonant frequency of the quartz wafer of the FPI sensor (0.19 MHz). However, it is not clearly described the relationship between these two equations.

5.       Moreover as far as can be understood from equations (1) and (2) the FPI needs that the ultrasonic waves have certain intensity and frequency. Therefore authors must discuss this point, and also describe with detail the characteristic of the generated ultrasonic waves and in general how the generator is operating (point 2).

6.       In line 150, authors stated that “The wavelength the FP fiber sensor was 1557.50 nm (At this wavelength, the slope of the waveform was bigger which means higher sensitivity).” But later, on line 193 it is mentioned that for the experimental measurements the tunable laser was set at 1565.7 nm. Here it is necessary to define why the laser was not set at the optimal wavelength (1557.50 nm).

7.       In line 213, it was stated “At 700 ºC, the spectrum of the FP fiber sensor had a big difference compared with that under room temperature as shown in Figure 7”. This sentence it is vague in general since did not describe the behavior of the FPI spectrum as the temperature is changing and therefore what it can be expected how this affect or how is related with the temperature sensing. Therefore I recommend to discuss and clarify this issue.

8.       In figure 8, are shown a couple of plots showing the FPI sensors output as a function of time. Here it is measure the time delay before the sensor start to oscillate. However can you describe if there is a relationship between the frequency of these oscillations and the frequency of the generated ultrasonic waves.

9.       In figure 9, I would like to recommend to add a figure showing the relationship between the “travel time” vs the temperature.

10.    In line 93, replace “black PDMS was coated on a glass slide” --> “a glass slide was coated with black PDMS”

11.    For clarity purposes, in line 110, I suggest to replace E = 7.2*1010 --> E = 7.2×1010

12.    In line 115 the sentence “E is Young’s modulus of quartz coverslip, E = 7.20*1010 Pa; μ is the Poisson ratio of quartz μ = 0.17; h is the thickness of the diaphragm h = 0.10 mm; d is the diameter of the diaphragm d = 2.54 mm.” can be deleted since these information was provided in page 2.

13.    In line 117, I suggest to rewrite “f00 could be calculated as 0.19 MHz.” --> “For our experiment f00 was calculated as 0.19 MHz.”

14.    Please replace, where necessary” ºC by ᵒC

Author Response

Please see the attached report notes.

Round 2

Reviewer 1 Report

Authors have addressed the concerns of the reviewer.

Author Response

We truly appreciate all the constructive comments and suggestions from the reviewer.

Reviewer 3 Report

Authors present the design of a system for temperature sensing. The system is composed by 2 stages, where the first one is an optical fiber ultrasonic generator and the second stage is a fiber the Fabry-Perot (FP) fiber sensor which monitors the ultrasonic waves. According to the authors the sensing system is able to detect the temperature of the air between the ultrasonic generator and the FPI sensor. Moreover, have attended in general the reviewer comments and have enhanced significantly the manuscript. Therefore I suggest to consider the paper for publication after some minor revisions.

Some specific aspects that I consider should be addressed by authors are the following.

1.       In line 112, authors mentioned that “After performing the Fourier transform, the bandwidth of the frequency range was at least 20 MHz”. Here I suggest to include as a Figure 2a the FFT spectrum of the signal to support and to clarify the point.

2.       In line 235, authors included a new sentence which I consider is important, however I think that it is just necessary to be more specific about the difference about the difference between the spectra of the two FPIs used. I guess that can be just spectral shift due to slight difference in thicknesses of the wafers/layers forming the FPI. But please shortly clarify this issue.

3.       In general the English edition must be carefully checked and corrected where necessary.

For instance:

In line 130: “I is the FP cavity length in μm, I = 3 μm” -->”where I is the FP cavity length (I = 3 μm)”

In line 130: “which meant, with” --> “which means that, with”

In line 171: “The wavelength the FP” --> “The wavelength of the FP”

In line 132: “(At these wavelengths,” --> “(between these wavelengths,”

Author Response

Please see the attached report notes.
